# Study on the Difference of Human Body Balance Stability Regulation Characteristics by Time-Frequency and Time-Domain Data Processing Methods

**DOI:** 10.3390/ijerph192114078

**Published:** 2022-10-28

**Authors:** Xinze Cui, Baosen Fu, Siqi Liu, Yuqi Cheng, Xin Wang, Tianyu Zhao

**Affiliations:** 1Department of Kinesiology, Shenyang Sport University, Shenyang 110115, China; 2School of General Education, Shenyang Sport University, Shenyang 110115, China; 3Shanghai Institute of Physical Education, Shanghai 200433, China; 4Key Laboratory of Structural Dynamics of Liaoning Province, College of Sciences, Northeastern University, Shenyang 110819, China

**Keywords:** overall stability index, wavelet transform, balance control ability

## Abstract

This paper aims to investigate the differences in standing balance control ability between freestyle skiing aerials athletes and ordinary graduate students by means of wavelet transform (WT) and the overall stability index (OSI) and to discover the characteristics of the postural control ability of dissimilar subjects and appropriate methods to assess the postural control ability of the human body. Research Methods: In all, 16 subjects were tested, including 8 from the Chinese national team who had won the world championships of freestyle skiing aerials, with 10+ years of training (age: 23 ± 23.1 years, Height: 176 ± 2.1 cm, and weight: 69 ± 3.5 kg), and 8 ordinary graduate students of Shenyang Institute of Physical Education (age: 22.6 ± 4.6 years, Height: 179 ± 3.3 cm, and weight: 73 ± 4.1 kg). When performing the tasks, the research subjects were required to stand on the steady support surface (with eyes closed and legs closed) for 30 s in each testing. The displacement data of the anteroposterior (AP) direction and the mediolateral (ML) direction of their centre of pressure (COP) were recorded. Then, WT and OSI were calculated. Two dissimilar methods were compared to analyse the characteristics of balance ability. Results: (1) The athletes’ WT values in the AP direction and the ML direction were concentrated in the interval of 22~30 s and 0–8 s, respectively, while the ordinary graduate students’ WT values in the AP direction and the ML direction were concentrated in the interval of 10~25 s and 0–7 s, respectively; (2) the WT values of the regular graduate students in the AP direction and the ML direction were higher than those of the athletes (*p* < 0.01); and (3) the OSI value in the AP direction of the athletes was higher than of the ordinary graduate students, while the OSI value in the ML direction of the athletes was lower than that of regular graduate students. Conclusion: Compared to the OSI, WT can analyse the characteristics of balance control ability more effectively. The COP displacement frequencies of the athletes and ordinary graduate students were concentrated in the low-frequency bands. The athletes had superior adjustment ability in an imbalanced state and could adjust to the best position without effort. In addition, the athletes had a stronger adaptive ability. In comparison, the ordinary graduate students had comparatively poor adaptive ability and weak adjustment ability in the imbalanced state, so it was difficult for them to attain the best angle after adjustment.

## 1. Introduction

The ability to control balance is the main component in positional control [1]. According to the conventional view, balance control is the automatic and reflexive procedure of action adjusted by the brain’s subcutaneous structure, which does not need a sense of control or input from attention resources [2]. However, a rising number of dual task studies show that the frontal and parietal lobes [3] are activated during the procedure of balance control, which necessitates the participation of attention resources and belongs to controlled processing. The realistic attention resource allocation of balance control is the foundation for integrating surrounding stimuli and completing action tasks, particularly in competitive sports with intense and complex environments [4]. 

As the primary assessment of balance capability, the displacement of foot COP in X direction and Y direction was recorded by the Biodex Balance System (BBS), and then the equivalent OSI was calculated by the equivalent formula. The OSI assesses the quality of balance capability by combining the tilt degree of the AP direction and ML direction [5]. The score is measured by the average deviation of the centre of gravity swing path around the zero point, using the unit ° [6]. According to one study, based on the results of five tests, the reliability of the OSI test results is as high as 85% [7].

The foot COP refers to the centre point of the whole pressure on the foot–ground contact surface. As a comprehensive variable, it reflects limb sway and posture balance control ability [8]. COP signal is unstable and nonlinear [9]. 

Linear methods assume that posture sway is steady when evaluating COP signal, so it can only analyse the change information of COP displacement, yet disregards the time structure information in the signal [6]. Consequently, it will cause the deviation of the COP time series and obscure the dynamic characteristics of COP data when the human body remains stable. Many more studies have shown that postural control is attained by visual, proprioceptive, and vestibular sensory organs, while the dynamic alterations of the postural control system cannot be assessed effectively if people only assess the swing amplitude and disregard the overall change by decreasing the change in postural control. Thus, higher evaluation methods are required to measure the dynamic changes in the postural control system [9]. The nonlinear time-domain and frequency-domain method, which is based on the concepts of chaos, fractals, and complexity [10], has been used to evaluate COP signals. In recent times, the majority of studies have utilised nonlinear time series analysis methods to analyse COP. Some researchers instead argue that complex postural control is more stable and adaptable [9]. 

WT can examine numerous different time scales at different points in time and offer pertinent information [11]. Continuous WT and discrete WT have been used to assess the motion trajectory of COP [12] and decompose the motion trajectory of COP in the time domain on multiple time scales [13]. By comparing the fast Fourier transform and WT, prior studies have revealed that the frequency domain range of COP displacement was 0–5 Hz. Additionally, disproportionate sampling frequency causes additional high-frequency noise, rather than reflecting the actual fluctuations in posture control [14]. It was established in one study that discrete WT could be used to analyse COP trajectories in the time domain on multiple time scales, but not enough decomposition conclusions were reached [15].

In this study, through OSI and WT, two dissimilar analysis methods were utilised to assess the balance control characteristics of outstanding athletes and ordinary graduate students under conditions of having closed eyes, closed legs, and a hard cushion. In addition, the right method applicable to the assessment of balance position control was analysed. It was clear that: (1) dissimilar subjects showed dissimilar characteristics in dissimilar time scales and (2) the athletes had more complex sway motion changes than regular graduate students. 

## 2. Materials and Methods

### 2.1. Research Objects

In all, 16 subjects were selected as research subjects and tested, including 8 first-class freestyle skiing aerials athletes from the Chinese national team with 10+ years of training (age: 23 ± 23.1 years, height: 176 ± 2.1 cm, and weight: 69 ± 3.5 kg) and 8 ordinary graduate students of Shenyang Institute of Physical Education (age: 22.6 ± 4.6 years, Height: 179 ± 3.3 cm, and weight: 73 ± 4.1 kg), according to the sample size of Gpower 3.1 software (Informer Technologies, Inc., Los Angeles, CA, USA). Each of the research subjects signed informed consent forms prior to performing physical activity tests at least three times per week. In addition, they all completed a questionnaire of their medical history to verify the lack of musculoskeletal, neurological, or other impairments that would affect their balance. The outcome of this investigation revealed that none of them had experienced any lower limb injury in the last 6 months. The experiment had the approval of the Ethics Committee of Shenyang Institute of Physical Education. 

### 2.2. Test Method

The test was conducted in the Sports Biomechanics Laboratory of Shenyang Sport University. COP movement throughout the standing procedure of subjects was recorded with a portable balance instrument (Humac Balance, American, sized at 65 cm × 40 cm). The pertinent parameters were analysed by software. At the sampling frequency of 100 Hz, the test was carried out in a quiet room with external interference and noise minimised. To ensure the subjects were relaxed and did not suffer from fatigue during the test, the test was arranged to be held on their adjusted rest day when they did not have training. Throughout the test, the subjects were asked to take off their footwear and stand on a 65 cm × 40 cm balance board (Humac Balance, American) “keeping their trunk as stable and motionless as possible”. The athletes were required to complete the balance test (with closed legs, closed eyes and hard cushion) on the stable plane for 30 s each time [11]. The subjects were asked to try and maintain their balance. Their COP change in both the AP and ML directions were recorded. Everyone remained quiet during the entire test process to ensure the subjects had a comparatively silent environment for testing and to decrease the interference of noise on their balance capability. According to the standard test scheme, the bottom of the second metatarsal bone of subjects’ feet was aligned with a mark on the floor. All subjects had to keep their eyes tightly closed and avoid any other movements. All subjects were given 30 s of sitting rest between the two tests. During the entire experiment, the subjects had to close their eyes, keep their hands at their sides and their bodies in a neutral position. 

### 2.3. Data Processing

#### 2.3.1. OSI 

All data were analysed by utilising SPSS 26.0 and tested through COP dissimilarities between groups. All the test levels were set to 0.05 as important dissimilarities. The diversities between dissimilar populations were identified by using a paired sample *t*-test. Results are expressed as mean (M) and standard deviation (SD). In addition, the effective size was calculated using Cohen’s D method. If the value was <0.2, it signified a small effect size; if the value was close to 0.5, it signified a moderate effect size; if the value was >0.8, it signified a large effect size. Between them, OSI is the overall stability index; APSI is the anteroposterior stability index; MLSI is the mediolateral stability index. SI signifies the stability of the human body in maintaining balance. The closer the SI comes to 0, the better the balance ability [13]:(1)$$\mathrm{APSI}=\sqrt{\frac{\sum {\left(0-y\right)}^{2}}{\#\mathrm{samples}}}$$
(2)$$\mathrm{MLSI}=\sqrt{\frac{\sum {\left(0-x\right)}^{2}}{\#\mathrm{samples}}}$$
(3)$$\mathrm{OSI}\;=\sqrt{\frac{\sum (0-x{)}^{2}+{\left(0-y\right)}^{2}}{\#\mathrm{samples}}}$$

#### 2.3.2. Wavelet Transform

The COP was recorded with a sampling frequency of 100 Hz and a sampling time of 30 s. The information estimated through the COP was completed in Matlab 2021b. 

WT is a signal analysis technique in the area of frequency research. From a mathematical perspective, WT is the wavelet convolution of the time series signal with dissimilar scales (*a*) and translation (*b*). The transform changes the real-time sequence signal *s*(*t*) into a two-dimensional real space with variables *a* and *b*. The wavelet coefficients (WC) at time scale *a* and time *b* are estimated using Equation (4) [9]:(4)$$\mathrm{WC}=\int_{-\infty }^{+\infty }s\left(t\right)\psi_{a,b}^{*}\left(t\right)dt,$$
where *s*(*t*) represents the time series signal; “*” represents the complex conjugate; and *ψ* is the wavelet function. The ‘child wavelet’ derived from the ‘mother wavelet’ is as shown in Equation (5):(5)$$\varphi_{a,b}\left(t\right)=\frac{1}{\sqrt{a}}\psi \left(\frac{t-b}{a}\right),\;$$

The above is a function of time scale “*a*”, and the mathematical similarity with the short-time Fourier transform (STFT) could be seen close to time “*b*”. Nevertheless, the results achieved by WT describe “*a*” and “*b*” in space rather than in time and frequency space. In the case of discrete wavelet transform (DWT) for signal digital processing, the values “*a*” and “*b*” are shown as in Equation (6):(6)$$a=2^{j}\;\Lambda \;b=2^{jk},\;$$
where *j* = 1…; *j* is the discrete level of time scale, *k* = 0…; the range of time scale *k* (*j*) is shown in Equation (7):(7)$$t_{j}=\left\{\frac{2^{2j-1}}{f_{s}},\frac{2^{2j}}{f_{s}}\right\},$$
where is the signal sampling frequency, and *k* (*j*) is represented by correlation (8):(8)$$k\left(j\right)=\mathrm{floor}\left(\frac{N-2^{j}}{2^{j}}\right),$$
where the floor function is rounded down to an integer by using Equation (4), and WT in the discrete space transforms the signal *s*(*t*) determined for the discrete value *t* into a two-dimensional signal *DWT* in the space of discrete level *j* and discrete position *k*.

At each discrete time level, *DWT* may be utilised to rebuild the time series signal at that particular level, thus producing the “detailed” time series signal—cD(*t*), as demonstrated in Equation (9):(9)$${\mathrm{cD}}_{j}\left(t\right)=\overset{K\left(j\right)}{\underset{k=0}{\sum }}DWT\left(j,k\right)\psi_{j,k}\left(t\right),$$

Equation (10) can be derived and constitutes an estimation of the time series signal at the specific level cA: (10)$${\mathrm{cA}}_{j}\left(t\right)=\sum_{k=0}^{K\left(j\right)}DWTap\left(j,k\right)\varphi_{j,k}\left(t\right),$$
where *φ* represents the direct scaling function *ψ* linked to the wavelet function; *DWTap*(*j*,*k*) represents the estimated value of DWT defined by Equation (11), being comparable to Equation (4):(11)$$DWTap\left(j,k\right)=\int_{-\infty }^{+\infty }s\left(t\right)\varphi_{j,k}\left(t\right).$$

## 3. Results

### 3.1. Comparison of WT Spectra of Different Subjects and Different Standing Directions

Table 1 shows the basic information of the subjects, Figure 1 shows the WT spectra diagram in the subject balance test. The frequency of excellent athletes is more concentrated in low frequency ranges and changes repeatedly. Ordinary graduate students are more discrete and change more often than excellent athletes. As can be seen, the energy of outstanding athletes in the AP direction was concentrated in the intervals of 0~5 s and 22~30 s, whilst that in the ML direction was concentrated in the interval of 0~8 s. In comparison, the energy in the AP direction of regular graduate students was mostly concentrated in the first 3 s of the test and in the interval of 10~25 s. Their 0~7 s swing in the ML direction was stronger than that in the AP direction, and there were two comparatively large energy vacillation zones. Modifications were made after 7 s, and there was long shaking period, which was mostly concentrated in the intervals of 0~15 s and 21~30 s in the test. 

### 3.2. Comparison of WT and OSI of Different Subjects and Different Standing Directions

Figure 2 shows the comparison between the WT of athletes and ordinary graduate students showed that the WT of athletes <the WT of ordinary graduate students in the AP direction (*p* < 0.01; d = 3.54), and the WT of athletes < the WT of regular graduate students in the ML direction (*p* < 0.01; d = 2.84). Thus, the WT values of athletes in both the AP and ML directions were less than those of ordinary graduate students. There were also major differences. The comparison between the OSI of athletes and regular graduate students showed that the OSI of athletes >the OSI of ordinary graduate students in the AP direction (*p* > 0.05; d = 1.02), and the OSI of athletes < the OSI of ordinary graduate students in the ML direction (*p* > 0.05; d = 0.92). There was no important dissimilarity in the AP and ML directions. In addition, for the WT of athletes, the WT in the AP direction < WT in the ML direction (*p* < 0.01; d = 2.73); for the ordinary graduate students, the WT in the AP direction > the WT in the ML direction (*p* < 0.05; d = 1.74). The energy value of athletes in the AP direction was smaller than that in the ML direction, but it was the reverse for the ordinary graduate students. Furthermore, for the OSI of athletes, the OSI in the AP direction < the OSI in the ML direction (*p* > 0.05; d = 0.94); for the ordinary graduate students, the OSI in the AP direction > the OSI in the ML direction (*p* < 0.01; d = 1.1). The OSI of athletes and ordinary graduate students in the AP direction was higher than that in the ML direction. 

## 4. Discussion

In this study, by comparing WT and OSI, the process most appropriate for the assessment of human body balance control capability was established and the characteristics of the balance control method were shown. Comparatively low wavelet energy presented by the human body was able to maintain its stability in time series and had excellent balance control capability. Comparatively large wavelet energy presented to the human body was unable to effectively maintain its stability and had poorer balance control capability. 

### 4.1. Change Characteristics of WT and OSI 

WT can analyse dissimilar signal components and evaluate the characteristics of each signal with the resolution matching its scale. The dynamic reaction of a structure is caused by the interference of operating and environmental conditions. Generally speaking, the length of high-frequency events is short, while low-frequency events are long. Recently, studies to observe health through comparative analysis of wavelet scale maps have been conducted [16].

The results (Figure 1) revealed that the COP swing frequencies of the athletes were mainly found in the low-frequency area. When athletes stood on the stable support surface, the peak energy of the AP direction materialised in 10~18 s, and the lowest value of fluctuation materialised in 30 s. The stable moment of the AP direction emerged in the interval of 1~6 s in the early test period. The peak value of energy in the ML direction emerged in 0~8 s, and the lowest value of fluctuation emerged in the interval of 20~22 s. The stable moment of the ML direction emerged in the interval of 10~25 s in the middle and later periods of the test. The swing change of the AP direction in the 30 s stable support surface test was steadier than that in the ML direction. When ordinary graduate students stood on the stable support surface, they had a greater swing change in the AP direction than in the ML direction. The peak value of energy in the AP direction emerged in the interval of 17~24 s, and the lowest value of fluctuation emerged in 20 s. In addition, the stable moment of the AP direction emerged in the interval of 15~25 s before and during the middle period of the test. The peak value of energy in the ML direction emerged in the interval of 5~10 s, and the lowest value of energy emerged in 15 s. The stable moment in the ML direction emerged in the interval of 10~20 s in the middle of the test. The swing was comparatively stable in the 30 s of the ML direction. The freestyle skiing aerials athletes had trained and competed on uneven snow for extended periods of time, so they were stable on both feet. In the early period of the test, the athletes were not adaptive, and they had a more noticeable swing with a greater energy value. As the athletes gradually adapted, the swing became smaller. There was still a comparatively small swing in the middle of the test. In order to adapt to imbalances after they closed their eyes, the athletes still needed to continually adjust their centre of gravity to attain a balanced state. There was still a comparatively large swing in the later period of the test because athletes suffered fatigue of the central nervous system after a long time of concentrated standing, integrated information slowly, and selected unsuitable strategies in the feedback for brake adjustment balance. The ordinary graduate students did not have any long-term systematic training, so they needed a far longer time to adapt to the support surface after closing their eyes. Additionally, some adjusted incorrectly and fell down when trying to adapt to the support surface. Relative to the athletes, the ordinary graduate students had a poorer ability to engage muscles in unbalanced situations, could not adjust their body posture subsequent to losing balance, and could not adjust to the most favourable state after adjustment. As a result, in the 30 s balance test, the central nervous system of ordinary graduate students became prematurely tired, causing errors in giving instructions to muscles, integrating information and other coping strategies. Consequently, our first hypothesis was valid.

### 4.2. The Change Characteristics of the Athletes and Healthy College Students in Different Directions

Athletes’ wavelet energy value in the AP direction was less than in the ML direction. Based on the comparison result, the OSI in the AP direction was higher than that in the ML direction. Additional research was conducted on their balance in the AP direction. The athletes had trained on snow for a long time, and they needed to attain concurrent taking-off and landing of their legs over the entire technical movement process of freestyle aerial skiing. In addition, during the process of sliding, to maintain balance control, they maintained stability solely by correcting the AP direction, and it was not easy for them to swing in the ML direction to maintain their own balance. When aerial skiing athletes stood on both feet, there were specific differences between their stability in the AP and ML directions. This is different from the findings of other projects [17]. By comparing the balance ability of ballet athletes and track and field athletes, it was established that track and field athletes had superior balance ability in the AP and ML directions, while ballet athletes had superior balance ability in the AP direction [18]. Comparison with the static one-leg standing of football players, basketball players, and healthy college students established that football players had less swing in the AP direction, but had a stronger ability to remain standing on one leg. In addition, studies revealed that athletes on the ice appeared to have directional and specific swing control. Due to the specificity of the sports, athletes who skated on ice had to control their balance by sliding in the AP direction. The skaters had less swing in the ML direction than most land sports athletes [19]. All skilled athletes had unique balance characteristics. Consequently, the conclusion was that wavelet energy precisely described the athletes’ balance changes in the 30 s interval in dissimilar directions, thus proving our second hypothesis.

Measured by WT and OSI, it was established that healthy college students had a greater swing in the AP direction than in the ML direction. Since the weight of the human body mostly falls on the appendage and metatarsal bone of the foot when standing, the bearing capacity of the appendage bone is superior to that of the metatarsal bone, and the arch chiefly bears the body weight. When the human body stands, its balance is chiefly maintained by the tightening of the triceps muscle of the calf [20]. Studies have shown that the shaking range of the body is primarily in the front and back direction in the standing position [21], which is identical to our research results.

### 4.3. Comparative Analysis of Different Analysis Methods

OSI assesses the human body’s balance ability by describing and recording the static posture. The displacement range of COP is measured in a static state and obtained by a two-dimensional time series platform. In addition, the COP trajectory [22] that maintained balance in the AP and ML directions was analysed. It was expressed as the weighted average of all the pressure of the contact between the human body and the ground area. According to studies conducted in more recent years, the balanced analysis ability of the weighted average of COP displacement is limited because the weighted average can only provide the magnitude information of COP displacement changes and disregards the temporal structure information of the COP signal [23]. Consequently, such methods cannot clarify the temporal changes caused by complex sensory-motor integration. As a result, it causes the deviation of the COP time series and obscures the characteristics of human changes. WT can precisely assess the dissimilarities of human movement and balance control capability [24], can respond to different time scales under diverse time characteristics [9], and can also show the postural control mechanism at dissimilar time scales [25]. Additionally, WT can not only analyse the energy of COP displacement in a specific time period to illustrate the present problems in posture control [26] but can also quantify the multi-resolution spectrum of COP displacement signal to depict the smoothness and diversity of the fractal structure of the signal [27]. WT is connected to the exact analysis of the low-frequency band of COP displacement and can analyse the energy changes in a precise time period to reveal the problems that exist in postural control and represent postural features [28]. Other studies have shown that the calculation method of WT was effectual in COP displacement in the low-frequency band and achieved the high precision design that other algorithms could not achieve in the same study [9], which was identical to the results established by this study. In addition, one study demonstrated that WT was appropriate for diseases connected to the identification and diagnosis of balance [19]. It has been discovered that WT can reveal the change in COP displacement that cannot be observed by conventional methods [29,30,31]. Studies have revealed that the increase in energy in the frequency band indicated that subjects were at risk of falling, which was similar to the results of this study, and the larger the WT, the worse the balance capability [9]. OSI is basically described via the mean and standard deviation, disregarding the change characteristics of the human body under time series. WT can better highlight the change rules of the human body by extracting the swing characteristics at different times for analysis. Consequently, compared with OSI, WT was more appropriate for analysing the information displayed when the balance capability was affected under dissimilar situations, so as to better comprehend the balance control capability.

## 5. Conclusions

According to the findings of this study, WT can successfully analyse the characteristics of balance control capability. The WT method can extract the time-frequency characteristics of balance control, preserve the low-frequency integrity in the process of human swing, and disclose more prospective features. The swing frequencies of COP amongst outstanding athletes and ordinary graduate students are concentrated in the low-frequency area, and outstanding athletes have more apparent characteristics. In the case of imbalance, outstanding athletes can adjust rapidly, keep their centre of gravity stable after adjustment, and also adapt more swiftly to a complex and changeable environment. In the imbalanced state, ordinary graduate students need a longer time to adjust and find it difficult to adjust to the paramount stable state after imbalances. The athletes showed better balance control ability in the AP direction than in the ML direction, while ordinary graduate students showed better balance control ability in the ML direction. Because of the specificity of this study and restricted samples in the experiment, it is essential to conduct additional research on more samples in the future.

## Figures and Tables

**Figure 1 ijerph-19-14078-f001:**
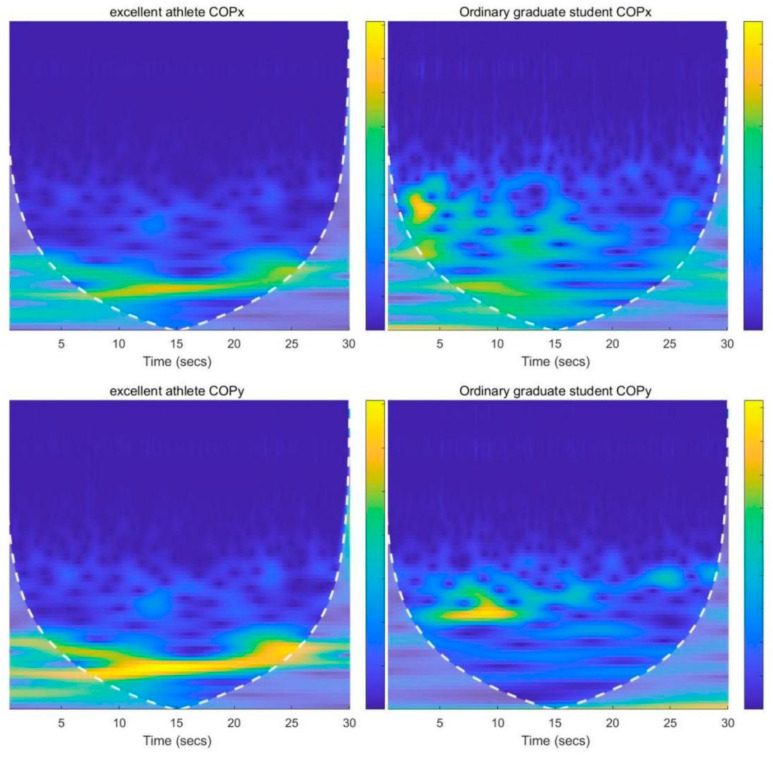
WT Multi-scale Spectra Diagram of COP for Different Subjects.

**Figure 2 ijerph-19-14078-f002:**
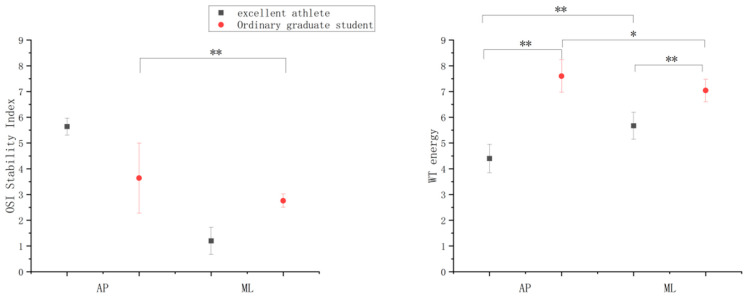
Comparison of significance between the WT and OSI of different subjects. Note: * indicates the difference between different subjects in different standing directions *p* < 0.05; ** indicates the difference between different subjects in different standing directions *p* < 0.01.

**Table 1 ijerph-19-14078-t001:** Basic information of subjects.

	Excellent Athletes			Ordinary Graduate Students		
	Height (cm)	Weight (kg)	Age (year)	Height (cm)	Weight (kg)	Age (year)
Male	176 ± 2.1	69 ± 3.5	23.2 ± 3.1	179 ± 3.3	73 ± 4.1	22.6 ± 4.6

## Data Availability

The data presented in this study are available from the corresponding author upon reasonable request.

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
