# Peer review of "Study on the Difference of Human Body Balance Stability Regulation Characteristics by Time-Frequency and Time-Domain Data Processing Methods"

_ijerph, 2022, doi:10.3390/ijerph192114078_

Round 1
Reviewer 1 Report
This manuscript compares two methods to assess standing postural control position and mentions the differences between freestyle skiers and postgraduate. The comparative analysis could be interesting if it were well justified, but with this wording and analysis it is not considered to be of relevant scientific interest. For example, the Wavelet transform has not been used in this field for years because there are more adequate methods and, therefore, to be an adequate topic, it should be justified with a more elaborate state of the art. Likewise, it is not detailed why the comparison between the selected sample groups is important. Due to the above and the most significant considerations set out below, it is considered that the study should be improved and rewritten.
Introduction
In general, the introduction needs to be rewritten because the description of human static equilibrium is a bit outdated. For example, it is mentioned that the CNS only controls two joints for postural control, when it is known to encompass more joints. Moreover, there is information that is generalized or modified from an original study. For example, lines 50 and 51 quote a study, which does not conclude the same as the authors have written in this manuscript. In fact, this statement is generalized to all humans, when the cited study collects a very specific sample and it should not be generalized. It is recommended to search for more current studies and improve the section.
In addition, the state of the art of the study is not justified. Why are the results sought important? Why has this topic been chosen? All this needs to be clarified.
Methods
The sampling method used and the methodology must be specified.
Has any ethics committee evaluated this study and given its approval? This must be specified in detail.
Why have 30 s tests been done? This must be justified, as there is a lot of literature on the subject.
The authors use a very small sample and compare the data using a parametric t-test. This causes a type II error because it has no robustness or external validity. To avoid this problem, it is recommended to obtain a larger sample, analyze the distribution of the variables and homoscedasticity. In this way, the most robust and valid test for this sample would be known.
Discussion and Conclusions
For all the above, especially in the methodology section, these sections should be rewritten and adapted to the results of the manuscript.
Minor revision
The entire document should be checked for grammar and formatting.
When using an abbreviation, such as COP or WT, you only need to specify it the first time. Thereafter, the abbreviation must always be used throughout the document.
Check table 1, because it has errors.
The number of the equations (indicated in parentheses) must be updated as they are erroneous.
Line 198 should be removed.
In the references section, some references have the initial number repeated.
I hope that these indications help you to improve the study.
Author Response
This manuscript compares two methods to assess standing postural control position and mentions the differences between freestyle skiers and postgraduate. The comparative analysis could be interesting if it were well justified, but with this wording and analysis it is not considered to be of relevant scientific interest. For example, the Wavelet transform has not been used in this field for years because there are more adequate methods and, therefore, to be an adequate topic, it should be justified with a more elaborate state of the art. Likewise, it is not detailed why the comparison between the selected sample groups is important. Due to the above and the most significant considerations set out below, it is considered that the study should be improved and rewritten.
- In general, the introduction needs to be rewritten because the description of human static equilibrium is a bit outdated. For example, it is mentioned that the CNS only controls two joints for postural control, when it is known to encompass more joints. Moreover, there is information that is generalized or modified from an original study. For example, lines 50 and 51 quote a study, which does not conclude the same as the authors have written in this manuscript. In fact, this statement is generalized to all humans, when the cited study collects a very specific sample and it should not be generalized. It is recommended to search for more current studies and improve the section.
In addition, the state of the art of the study is not justified. Why are the results sought important? Why has this topic been chosen? All this needs to be clarified.
Reply:The introduction has been modified and rewritten
- The sampling method used and the methodology must be specified.
Reply:Inclusion criteria have been added
- Has any ethics committee evaluated this study and given its approval? This must be specified in detail.
Reply:Has been added to the Ethics Committee instructions
- Why have 30 s tests been done? This must be justified, as there is a lot of literature on the subject.
Reply:References were added after the testing
- The authors use a very small sample and compare the data using a parametric t-test. This causes a type II error because it has no robustness or external validity. To avoid this problem, it is recommended to obtain a larger sample, analyze the distribution of the variables and homoscedasticity. In this way, the most robust and valid test for this sample would be known.
Reply:Dual validation of the Cohen effect with the t-test has been used
- For all the above, especially in the methodology section, these sections should be rewritten and adapted to the results of the manuscript.
Reply:To modify the discussion and conclusions
- The entire document should be checked for grammar and formatting.
Reply:Has been modified
- When using an abbreviation, such as COP or WT, you only need to specify it the first time. Thereafter, the abbreviation must always be used throughout the document.
Reply:Has been modified
- Check table 1, because it has errors.
Reply:Has been modified
- The number of the equations (indicated in parentheses) must be updated as they are erroneous.
Reply:Has been modified
- Line 198 should be removed.
Reply:Has been modified
- In the references section, some references have the initial number repeated.
Reply:Has been modified
Reviewer 2 Report
ijerph-1877565_review
Title: A comparative study on analyzing human balance control ability based on wavelet transform and OSI stability
Comments and Suggestions for Authors
Dear authors,
I have carefully read your paper that focused on evaluating the characteristics of balance control between elite athletes and ordinary people from two different analysis methods, in addition to determining which of these two methods could be more adequate to evaluate postural control of balance.
In my opinion, although your study included an evaluation to test a suitable method to assess balance control ability, the results shown are insufficient to draw a definite conclusion. I believe that more research would be necessary in this field with a larger number of participants, to really test the usefulness of this method in the exploration of both the general population and elite athletes from other fields. This could be useful to assist exercise and health professionals and researchers in planning, promoting and implementing complex interventions for postural control training with specific exercise programs.
In general, the manuscript is understandable. I found some issues in abstract, materials and methods, results and conclusions sections that should be addressed to improve the paper, in my opinion.
Specific comments:
Title:
- Page 1, lines 2-3: In think the title could be more specific to improve its dissemination and visibility. I suggest not use abbreviations or acronyms; readers may not know the meaning of OSI acronym.
Abstract:
- Page 1, line 11: Add a space between Background: By. There are missing or excess spaces throughout the abstract and manuscript, please check them.
- Page 1, lines 13-14: Please review this sentence: “and chooses a more suitable one for evaluating the human body Methods of posture control. Research”.
- Page 1, lines 15-16: I suggest you to add this information about the participants in the results section.
- Page 1, line 19. I suggest that just as you have done with the terms COP, AP and ML, first name the meaning of the term OSI.
Introduction
- Page 2, lines 57, 67: You have already used the acronym for COP and WT, once an acronym is used for a word, this acronym should be used in the rest of the manuscript so you should use that acronym every time this term appears throughout the text from then on. Please check them throughout the manuscript.
- Page 2, lines 63-65. What does the acronym OSI stand for? Overall stability index? This term is central to your manuscript, please clarify.
Materials and methods
- Page 2, lines 87-89, 94: I suggest you to add the information about the participants and table 1 in the section of results. It would be more appropriate to add this information in the results section.
- Page 2, lines 87-93. Please, could you add information about the period of time and places in which the data were recorded (i.e.: university, hospital…)?
- Page 2, lines 87-93: What inclusion or exclusion criteria were established to select the participants in the study?
- Page 2, line 90: You mentioned that informed consent was obtained from all participants. Was the study conducted in accordance with the Helsinki declaration? If so, please add this information.
- Page 3, line 108: In this line you mentioned the terms: “anteroposterior (arteriosclerosis, AP) and medial-lateral (multilateral, ML)”. I think the words arteriosclerosis and multilateral are a grammar mistakes, please check it.
- Page 3, lines 110, 111, 112. In these lines you describe how you performed the test. You repeat the term “athletes “all the time. I suggest you change it to “participants” or “subjects” since you are describing how all study participants performed the test, not just the athletes. If different protocols were used for each group they should be described. Please clarify.
- Page 3, line 119: You mentioned “results are expressed as mean standard deviation (MSD)”. I suggest change to “mean ± standard deviation (SD)”.
- Page 4, lines 133, 142, 144, 146, 152, 153. The numbering of the equations is incorrect from line 133, this would be equation 3, and so on in the rest of the lines, equation 4, 5, 6....I think it is a grammar mistake, please check it.
Results
- Pages 5-6, line 113: Figures 1 and 2 are really nice, congratulations. Please, check that all the words and terms that appear in the description of the figures and legends are in English and not in another language.
Discussion
- Page 6, lines 192-196. in this paragraph you mention the main findings of your study, it is not necessary to add the reference.
- You have not described the limitations of your study; please add them before the conclusions section.
Conclusions
Page 8, lines 281-293: I suggest reformulating your conclusions in a more careful way. Your results are insufficient to make a definitive statement, for example, you could use: "The results of the study suggested that...” Also, the conclusions must respond to the objectives of the study and not be limited to sharing the results of the study.
I hope that my comments could help to improve the paper.
Author Response
In my opinion, although your study included an evaluation to test a suitable method to assess balance control ability, the results shown are insufficient to draw a definite conclusion. I believe that more research would be necessary in this field with a larger number of participants, to really test the usefulness of this method in the exploration of both the general population and elite athletes from other fields. This could be useful to assist exercise and health professionals and researchers in planning, promoting and implementing complex interventions for postural control training with specific exercise programs.
In general, the manuscript is understandable. I found some issues in abstract, materials and methods, results and conclusions sections that should be addressed to improve the paper, in my opinion.
- Page 1, lines 2-3: In think the title could be more specific to improve its dissemination and visibility. I suggest not use abbreviations or acronyms; readers may not know the meaning of OSI acronym.
Reply:The title has been modified
Abstract:
- Page 1, line 11: Add a space between Background: By. There are missing or excess spaces throughout the abstract and manuscript, please check them.
Reply:The title has been modified
- Page 1, lines 13-14: Please review this sentence: “and chooses a more suitable one for evaluating the human body Methods of posture control. Research”.
Reply:The title has been modified
- Page 1, lines 15-16: I suggest you to add this information about the participants in the results section.
Reply:The title has been modified
- Page 1, line 19. I suggest that just as you have done with the terms COP, AP and ML, first name the meaning of the term OSI.
Reply:The title has been modified
Introduction
- Page 2, lines 57, 67: You have already used the acronym for COP and WT, once an acronym is used for a word, this acronym should be used in the rest of the manuscript so you should use that acronym every time this term appears throughout the text from then on. Please check them throughout the manuscript.
Reply:The title has been modified
- Page 2, lines 63-65. What does the acronym OSI stand for? Overall stability index? This term is central to your manuscript, please clarify.
Reply:An introduction to the OSI stability index has been added in the introduction
Materials and methods
- Page 2, lines 87-89, 94: I suggest you to add the information about the participants and table 1 in the section of results. It would be more appropriate to add this information in the results section.
Reply:The Table 1 information has been modified
- Page 2, lines 87-93. Please, could you add information about the period of time and places in which the data were recorded (i.e.: university, hospital…)?
Reply:Has been added in the experimental process
- Page 2, lines 87-93: What inclusion or exclusion criteria were established to select the participants in the study?
Reply:Has been added in the experimental process
- Page 2, line 90: You mentioned that informed consent was obtained from all participants. Was the study conducted in accordance with the Helsinki declaration? If so, please add this information.
Reply:All the subjects signed an informed consent form
- Page 3, line 108: In this line you mentioned the terms: “anteroposterior (arteriosclerosis, AP) and medial-lateral (multilateral, ML)”. I think the words arteriosclerosis and multilateral are a grammar mistakes, please check it.
Reply:Modification has been made
- Page 3, lines 110, 111, 112. In these lines you describe how you performed the test. You repeat the term “athletes “all the time. I suggest you change it to “participants” or “subjects” since you are describing how all study participants performed the test, not just the athletes. If different protocols were used for each group they should be described. Please clarify.
Reply:Modification has been made
- Page 3, line 119: You mentioned “results are expressed as mean standard deviation (MSD)”. I suggest change to “mean ± standard deviation (SD)”.
Reply:Modification has been made
- Page 4, lines 133, 142, 144, 146, 152, 153. The numbering of the equations is incorrect from line 133, this would be equation 3, and so on in the rest of the lines, equation 4, 5, 6....I think it is a grammar mistake, please check it.
Reply:Modification has been made
Results
- Pages 5-6, line 113: Figures 1 and 2 are really nice, congratulations. Please, check that all the words and terms that appear in the description of the figures and legends are in English and not in another language.
Reply:Already edited pictures from new
Discussion
- Page 6, lines 192-196. in this paragraph you mention the main findings of your study, it is not necessary to add the reference.
Reply:Added in the article
- You have not described the limitations of your study; please add them before the conclusions section.
Reply:Added in the article
Conclusions
- Page 8, lines 281-293: I suggest reformulating your conclusions in a more careful way. Your results are insufficient to make a definitive statement, for example, you could use: "The results of the study suggested that...” Also, the conclusions must respond to the objectives of the study and not be limited to sharing the results of the study.
Reply:The conclusions have been revised

Reviewer 3 Report
I really appreciate the opportunity to review this manuscript entitled “A comparative study on analyzing human balance control ability based on wavelet transform and OSI stability.” This is important to assess balance in this population. I remark some issues (most of them in methods) in order to improve the quality of this manuscript.
Tittle should be clearer, what is OSI? It should be avoid abbreviations in tittle and abstract. The abstract is clear but it is important to explain the analysis of the variables, a sex analysis was done, did authors find differences between sexes? Introduction was well structure and shows the necessity for this research. The aim of the paper is clear at the end of the introduction.
At the methods section, there are some questions that should be review. Which are the inclusion and exclusion criteria? Table 1 shows information that should be in Results. In the test methods, is it the same protocol for athletes and students? It should be explained. In data processing OSI stability index should be explained with more details.
About results. Figure 1 contains four images…which are from athletes and which emerge from students? Figure 2 should be translated into English. Discussion summarize and explain in a good way the finding but, from my point of view it would be interesting to discuss differences between sexes. What about a follow-up? Conclusions were correct but can be more concise.
Author Response
I really appreciate the opportunity to review this manuscript entitled “A comparative study on analyzing human balance control ability based on wavelet transform and OSI stability.” This is important to assess balance in this population. I remark some issues (most of them in methods) in order to improve the quality of this manuscript.
- Tittle should be clearer, what is OSI? It should be avoid abbreviations in tittle and abstract. The abstract is clear but it is important to explain the analysis of the variables, a sex analysis was done, did authors find differences between sexes? Introduction was well structure and shows the necessity for this research. The aim of the paper is clear at the end of the introduction.
Reply:1. . The title has been changed2. Gender has been modified
- At the methods section, there are some questions that should be review. Which are the inclusion and exclusion criteria? Table 1 shows information that should be in Results. In the test methods, is it the same protocol for athletes and students? It should be explained. In data processing OSI stability index should be explained with more details.
Reply:1.Inclusion exclusion criteria have been added.2.Figure 1 Information has been modified3.The OSI stability index has been explained in the article
- About results. Figure 1 contains four images…which are from athletes and which emerge from students? Figure 2 should be translated into English. Discussion summarize and explain in a good way the finding but, from my point of view it would be interesting to discuss differences between sexes. What about a follow-up? Conclusions were correct but can be more concise.
Reply:1.. The picture has been modified 2.The conclusions have been modified

Round 2
Reviewer 1 Report
The authors have made a major revision, significantly improving the manuscript. However, the results of the research do not provide relevant information to science, since previous articles have obtained the same conclusions. Still, the article has no errors and could be published.
Author Response
- The authors have made a major revision, significantly improving the manuscript. However, the results of the research do not provide relevant information to science, since previous articles have obtained the same conclusions. Still, the article has no errors and could be published.
Reply:Has enriched the research results
Reviewer 2 Report
ijerph-1877565_2review
Title: Study on the Difference of Human Body Balance Stability Regulation Characteristics by Time-frequency and Time-domain Data Processing Methods
Comments for Authors
Dear authors,
I was glad to have the opportunity to review the new version of your manuscript.
As I already mentioned this article studies the characteristics of balance control between elite athletes and ordinary people from two different analysis methods, in addition authors aim to determining which of these two methods could be more adequate to evaluate postural control of balance.
In my opinion, although you have made efforts to improve the manuscript, you have not responded positively to the suggestions for improvement made, on many occasions simply removing the text or information. For example you have eliminated Table 1
I would like to comment on some minor issues that could be addressed to improve the document, in my opinion.
Specific comments:
- Page 1, lines 15-16: I suggest you to add this information about the participants in the results section.
Reply:The title has been modified
New Comments: You have not responded to my suggestion
Introduction
- Page 2, lines 63-65. What does the acronym OSI stand for? Overall stability index? This term is central to your manuscript, please clarify.
Reply: An introduction to the OSI stability index has been added in the introduction
New Comments: You have not responded to my suggestion
Materials and methods
- Page 2, lines 87-89, 94: I suggest you to add the information about the participants and table 1 in the section of results. It would be more appropriate to add this information in the results section.
Reply:The Table 1 information has been modified
New Comments: You have not responded to my suggestion. You have removed table 1, my suggestion was to put it in the results section
- Page 2, lines 87-93. Please, could you add information about the period of time and places in which the data were recorded (i.e.: university, hospital…)?
Reply:Has been added in the experimental process
New Comments: You have not responded to my suggestion
- Page 2, lines 87-93: What inclusion or exclusion criteria were established to select the participants in the study?
Reply:Has been added in the experimental process
New Comments: You have not responded to my suggestion
- Page 3, line 119: You mentioned “results are expressed as mean standard deviation (MSD)”. I suggest change to “mean ± standard deviation (SD)”.
Reply:Modification has been made
New Comments: You have not responded to my suggestion
- Page 4, lines 133, 142, 144, 146, 152, 153. The numbering of the equations is incorrect from line 133, this would be equation 3, and so on in the rest of the lines, equation 4, 5, 6....I think it is a grammar mistake, please check it. No lo ha modificado en elt exto
Reply: Modification has been made
New Comments: Errors keep appearing in the numbering, please check them
Discussion
- You have not described the limitations of your study; please add them before the conclusions section.
Reply: Added in the article
New Comments: You have not responded to my suggestion
Author Response
- Page 1, lines 15-16: I suggest you to add this information about the participants in the results section.
Reply:The title has been modified,marked in the text
- Page 2, lines 63-65. What does the acronym OSI stand for? Overall stability index? This term is central to your manuscript, please clarify.
Reply: An introduction to the OSI stability index has been added in the introduction,marked in the text
- Page 2, lines 87-89, 94: I suggest you to add the information about the participants and table 1 in the section of results. It would be more appropriate to add this information in the results section.
Reply:The Table 1 information has been modified,marked in the text
- Page 2, lines 87-93. Please, could you add information about the period of time and places in which the data were recorded (i.e.: university, hospital…)?
Reply:Has been added in the experimental process,marked in the text
- Page 2, lines 87-93: What inclusion or exclusion criteria were established to select the participants in the study?
Reply:Has been added in the experimental process,marked in the text
- Page 3, line 119: You mentioned “results are expressed as mean standard deviation (MSD)”. I suggest change to “mean ± standard deviation (SD)”.
Reply:Modification has been made,marked in the text
- Page 4, lines 133, 142, 144, 146, 152, 153. The numbering of the equations is incorrect from line 133, this would be equation 3, and so on in the rest of the lines, equation 4, 5, 6....I think it is a grammar mistake, please check it. No lo ha modificado en elt exto
Reply: Modification has been made,marked in the text
- You have not described the limitations of your study; please add them before the conclusions section.
Reply: Added in the article,marked in the text
Reviewer 3 Report
I really appreciate the opportunity to review again this manuscript entitled “A comparative study on analyzing human balance control ability based on wavelet transform and OSI stability.” This is important to assess balance in this population. I remark some issues in order to improve the quality of this manuscript.
Tittle is clearer now. Introduction was well structure and shows the necessity for this research. The aim of the paper is clear at the end of the introduction.
At the methods section, there are some questions that should be review. Inclusion and exclusion criteria should be clearer and more explicit. Table 1 should be place in Results, where is now?
About results. Figure 1 contains four images…which are from athletes and which emerge from students? Now, it is clear. Figure 2 has been translated into English. Discussion summarize and explain in a good way the finding but, but in general, the text is difficult to rear with all the changes in red and highlighted and underlined.
Conclusions were correct.
Author Response
- At the methods section, there are some questions that should be review. Inclusion and exclusion criteria should be clearer and more explicit. Table 1 should be place in Results, where is now?
Reply:The inclusion criteria have been revised, and Table 1 is placed in the results of the study
- About results. Figure 1 contains four images…which are from athletes and which emerge from students? Now, it is clear. Figure 2 has been translated into English. Discussion summarize and explain in a good way the finding but, but in general, the text is difficult to rear with all the changes in red and highlighted and underlined.
Reply:marked in the text